# Love as Consolidation of the Self-in-the-World: Martin d'Arcy's Speculation on Love as a Metaphysical Supplement to Phenomenology

**Smilen Markov** [1,2]

1   Theological Faculty, The University of Veliko Tarnovo, 5003 Veliko Tarnovo, Bulgaria;
    smilenmarkov@gmail.com
2   The House of St. Gregory and St. Macrina, Oxford OX26LU, UK

**Abstract:** In his seminal work *The Mind and the Heart of Love. A Study in Eros and Agape*, Martin d'Arcy shows that self-sacrificial love (agape) and desire (eros) express the mystery of selfhood. Using the method of phenomenology, he demonstrates that eros and agape encompass a range of affectations, emotions and existential modes. All these make sense when seen as stages in the process of self-giving. Thus, eros and agape do not pertain to two opposing aspects of the soul. Rather, they are modes of manifestation of the entire person. In answering to the agapeic love of God, human agapeic love comes to a state which reason cannot grasp. At this point the erotic impulse steps in in order for the human soul to take the path of unknowing. Through this interplay the true hierarchy of being is perceived and the human person enters into loving exchange with the world. This happens within a three-tier process of loving knowledge whose structure is similar to the model of self-knowledge developed in the early Byzantine theological compendium *Corpus dionysiacum*.

**Keywords:** love; eros; agape; phenomenology; Christianity; hierarchy

## 1. Introduction

In his seminal work *The Mind and the Heart of Love. A Study in Eros and Agape*, Martin d'Arcy affirms that true love is only possible as self-giving. It is in self-giving that we experience love fully and creatively. No loving relation can be established among objects and lovers who do not give themselves entirely in fact objectify the beloved person. The question d'Arcy wants to answer is whether all types and layers of love are capable of partaking in self-giving or human love should undergo transformation in order to abandon the impulse to possess the Beloved. In order to explicate the aspects of possessive and self-sacrificial love, d'Arcy looks at the ramifications of the motives of erotic and agapeic love throughout intellectual history. Since the dawn of Christianity, the antinomy between possession and self-sacrifice permeates the different discourses on love, but it is specifically addressed when the relation between human love and divine love is at stake.

This analysis raises questions about the ontological status of the human person, the boundaries of selfhood, the nature of interpersonality and the capabilities of humans to know God and to relate with him. Instead of developing a metaphysics of love, d'Arcy focuses on certain topics which mark the historical trajectory of the discourse on love throughout the history of ideas. He uses as a starting point the interpretations of Denis de Rougemont and Anders Nygren of erotic and agapeic love. Both these thinkers assert the juxtaposition of eros and agape. The eros is a loving desire which ends up in the union with the Other, whereas agape is self-sacrificial love which amounts to self-giving. When these impulses reach to their extremes the erotic desire results in devouring of the Other and agape brings the human subject to self-destruction.

Rougemont and Nygren take different paths in examining love's relation to human knowledge. Whereas for Rougemont the eros is a blind destructive desire of the human

soul in need of cultivation by the rational, ethically balanced and pragmatic agape, Nygren sees the latter as a new existential impulse, introduced to human existence by Christ, which overcomes the erotic narcissism of human reason. D'Arcy attempts to subvert both these approaches and examines the different forms and manifestations of love from the perspective of Medieval Christian philosophy which conceptualizes love as a cognitive power par excellence.

D'Arcy is not eager to solve the antinomy of eros–agape, certainly not before he has studied its various implications for theological anthropology. Through historio-philosophical and historio-theological excurses, d'Arcy constructs a conceptual framework in which he offers a Christian phenomenological model of the experience of love. The commensurability between Christian anthropology and modern phenomenology was examined by d'Arcy in his brilliant monograph *Facing God*. In this work d'Arcy shows the potential of the phenomenological method to express the mystery of selfhood without superimposing any normative explanatory model on the dynamics of the person's life. It is in *Facing God* that d'Arcy comes to the conclusion that love constitutes the ontological core of the self. In his earlier work *The Lion and the Unicorn*, he shows that it is only through the ambiguities of love that one can achieve authentic interpersonal communion and freedom to accept the gift of God's indwelling in the human person.

## 2. Literature Review

The philosophical oeuvre of d'Arcy has rarely been the focus of scholarly research. One of the few monographs dedicated to him is *Martin d'Arcy: Philosopher of Christian Love* by J. A. Sire (1997). It examines his biography and intellectual career. Sire places the concept of love at the centre of d'Arcy's philosophy and theology. He underlines that d'Arcy's argument in *The mind and the Heart of Love* reflects the post-Lutheran context of polarization of human personality between passion and reason, *eros* and *agape*. The author regrets that d'Arcy has paid excessive attention to the dichotomy of *eros* vs. *agape* which had been propounded, in post-Lutheran context, by Anders Nygren and, partly, by Denis de Rougemont (1983): "he used a sledge-hammer to show that a nut is cracked" (Sire 1997, *Op. cit.*, 114).

For Sire the core of d'Arcy's argument is the concept of selfhood. Selfhood is a mystery that reveals divine being in the realm of created being; the self is "a unique reflexion of God's glory" (Sire 1997, *Op. cit.*, 114). In Sire's reading, d'Arcy's foundational concept of the self is countered by a conceptual anti-climax, namely the perplexing dichotomy *essential* vs. *existential* self, *eros* corresponding to the essential order and *agape*—to the existential one. The essential self seeks perfection through the existential self. According to Sire, d'Arcy leaves unsolved the problem of the integrity of the loving self (Sire 1997, *Op. cit.*, 115). It would seem that the identity of the self is guaranteed by the exchange between the loving impulses of the essential and the existential self; a connection characterised by friendly love, philia. The friendly interplay between *eros* and *agape* implies transfiguring human love into divine. At stake is a revelation of God within the human being (Sire 1997, *Op. cit.*, 116). However, in Sire's view, d'Arcy has failed to elaborate the theological implications of this divine presence.

In his monograph *The Spirit and the Forms of Love* (Williams 1981), Daniel-Day Williams dedicated one chapter to d'Arcy's concept of love. D'Arcy is placed among other modern thinkers, such as Albert Schweizer and Reinhold Niebuhr, who try to modify the traditional Christian concept of love. In his take on love, d'Arcy, claims Williams, oscillates between the Augustinian and the Thomistic tradition. In the Augustinian model, God is both the source and the ultimate object of truthful love. Thus, by transcending the boundaries of the self and uniting with God, human love has to be transfigured into devotion to God. According to the Thomistic approach, a human being imitates God's love by seeking itself and its own perfection. By deconstructing the simplistic normative juxtaposition between self-sacrificial and passionate love (and sexual desire, in particular) d'Arcy's speculation aims at integrating "the complex and dynamic view" of selfhood which has

emerged in modern psychology into the Christian theological discourse. Williams tries to show how this is achieved. D'Arcy's identification of *eros* and *agape* with the masculine and feminine principles is seen as symptomatic, as it proves the complementarity of the components of the life of the self. The traditional hierarchy of passionate and self-sacrificial love is substituted by the model of gradual development and expression of the powers of selfhood[1]. Williams demonstrates that d'Arcy transfers physiological and psychological paradigms into a philosophical discourse by using the strategies developed by modern existentialism, and more specifically by Hunter Guthrie. The latter interprets the human ego as a self-subsistent entity (essence) who, by virtue of its ontological principle, is seeking the Absolute to which it can give itself[2].

In this respect, Williams gives a much more in-depth interpretation of the dichotomy of *eros* vs. *agape*. He notes that, according to d'Arcy, the interplay of *eros* and *agape* in making and expressing the "I" is primarily of epistemological nature. The epistemological path of love comprises three layers: knowledge, ignorance and higher mode of knowledge. He quotes d'Arcy: "To be a person is to be essentially in search of a person. Love presupposes *knowledge*, but it can do to some degree without it; what it needs is the living and actual being itself"[3]. Here Williams detects a key element in d'Arcy's concept which has to do with the tradition of mystical theology and is indebted to Pseudo-Dionysius' model of apophatic and cataphatic predication of God. According to Williams, there is a lacuna in d'Arcy's understanding of divine love and its engaging in self-giving. D'Arcy's notion of love as a search of a person (cf. D'Arcy 1947, p. 41) seems incommensurate with divine love, as God is a perfect being and does not seek self-fulfillment. If the essence of divine love is to seek communion with creation, there must be some ontological model demonstrating this search of God for love which does not imply any essential change in God. I will claim that such a model is implied by d'Arcy and is congenial to Pseudo-Donysius' model of participation of created being in divine life.

I will maintain that the antinomies identified by Sire and the epistemological perspective reconstructed by Williams constitute an anthropological foundation for this personalized self-identification with the world. The archaeology of structures and perceptions of love, including the historical overview of the different concepts of love, is in fact a reconstruction of the process of dialogical personal knowledge. I will refer to Merleau-Ponty as the source of d'Arcy's systematic framework when clarifying the dialectics of love and its roots in the Christian mystical tradition dating back to Pseudo-Dionysius the Areopagite.

In this article d'Arcy's concept of love in *The Mind and the Heart* is read retrospectively through the phenomenological model of *Facing God*.

## 3. The Dichotomies of Love

D'Arcy denies the popular identification of *eros* with desire and *agape* with the active self-sacrificial disposition towards the beloved. Both these aspects, asserts d'Arcy, include active and passive vectors. He points out that *agape* entails not only self-sacrifice, but desire as well, just as eros implies not only possessing but also self-giving. For d'Arcy the distinction between desire and self-giving is not based on the differentiation between activity and passivity; it exemplifies two aspects of the self in relation to the beloved God. The eros is the indwelling of the person in the love of God, whereas the agape is the response of the human love towards divine love.

The human love to God is not merely a faculty or a one-off inclination. This is a *love story*, which has its source in the *agape* of God and corresponds to the *agape* of the soul lifted above itself. The detection of God's agapeic love and the response to it constitute faith. This is a transformative experience: not only for the lover, but for the world the lover dwells in. Faith is not just a passionate and instantaneous union. It is a long and painstaking process; it is a path into darkness. This darkness has nothing to do with the unconscious or the dark world of romance. As d'Arcy puts it, this is a darkness, which is due to excess of light. This motif is borrowed from the apophatic terminology of Pseudo-Dionysius the Areopagite

([Areopagita 1990b](#), [2014](#)). To come through it, one needs to follow the voice of the living God, and it is reason that could hear this voice ([D'Arcy 1947](#)).

In its agapeic upward movement, the spiritual anima comes to a state which reason cannot understand. This is because God is beyond any rational grasping. At this point the erotic impulse steps in. So, the anima must take the path of unknowing whereby the desire of the anima may turn into frenzy. At this state the integrity of the self is at risk. D'Arcy makes recourse to the symbolic interpretation of the masculine and feminine principle in the human person. At stake is not a symbolic exemplification of a specific mystical state of mind. Said symbols articulate the existential experience of self-knowledge through love. The two principles may alter their mutual relations, but they always stay together. Reason is driven by the animus (the male principle) and the erotic desire. The passionate anima, on the other hand, is not a merely passive agent, since it produces the impulse which leads the self to the abyss of unknowing. It turns out that the two members of the original antinomy of love are themselves antinomic and intricate. D'Arcy deconstructs a classical motif of the dual structure of the human being, dating back to Plato. In Plato's normative ethical model, the "inner man" of reason has to tame the capricious beasts of volition and anger[4]. D'Arcy offers a holistic and dynamic model of the antinomies of love.

## 4. The Dialectic of Love

The impulses of *agape* and *eros*, each of them containing antinomic aspects, are reconciled in faith, seen as union with the Word of God. The erotic aspect of religious experience is not confined to the moments of repose when we are detached from all practical aspects of our existence. D'Arcy is careful not to reduce the connotation of this reconciliation to overcoming of a physiological deficit or healing of a psychological trauma. Said aspects of love apply to the existence of men before God, that is to say his model of the interplay between eros and agape corresponds to an existential metaphysical concept of the experience of God.

If the agapeic desire of the anima is not counterbalanced by the eroticism of the animus, the anima passes through a period of romance. It aims at self-transcending in order to reach the beloved one. However, eventually this romance ends with the anima wanting to dissolve completely into nothingness. It starts craving for oblivion. Any acting of the self in which the latter is an instrument of the self-destructing anima is catastrophic. The anima has to listen to what the animus or the reason is telling her. Yielding to its own desires the anima imperils the immortal self. However, the animus is also a force that can lead to the destruction of the self. It cannot help wanting to grasp the reality—to make it its own and thus to form a body of knowledge. In other words, erotic desire leads to ego-centrism, the latter closing the person for any meaningful communion in love.

The way out of this conundrum is when the anima consciously strives to reach God, and not just any object of agapeic desire. This is the way of mystical contemplation. The contemplation of God is so sublime a state that the anima cannot but turn to the animus in order to confirm this experience within the personal being, i.e., within the human existence. In order to achieve this, the anima needs the animus. Thus, the anima would engage the animus in the act of love. In the face of the unfathomable God, the animus would be prevented from dictatorially imposing its own power over the reality of love. This is a model for meaningful synergy between anima and animus.

Although the desire for the Absolute is noble and has proved fruitful for many, d'Arcy warns that no way of life is more open to delusion, and many seem to confuse the means with the end. The risk here is that the communion with the living and bountiful God is sought not within the framework of the personal being. It is very easy to confuse the fervent desire for a union with the union itself. Additionally, the risk of delusion is connected to the way one expresses this experience. The human subject is turned into an object, which is impregnated with the divine life. However, thus the personal dimension of being is drained out both from the lover and from God. Love between persons precludes the reduction or the annihilation of the other. Love between persons means that each wants the other to be

more himself (D'Arcy 1947, p. 166). This is true on the level of love between human beings, as well as between a human person and God.

D'Arcy points out that the destructive tendencies of the self-sacrificing *anima* and self-regarding *animus* are based on a deeper uncertainty in the existence of the self. He carries out a historical overview of the concepts of human existence since Descartes and tries to explain the position of the various authors through the antinomy 'anima-animus': "The animus tends to make itself the measure of reality. . . . It resolves all that exists into its own essence, as if human thought and thing were correlative and, at the end, identical." (D'Arcy 1947, p. 256). In this sense, the projects of Kant and Hegel are seen as corrections to Descartes' rational absolutism. Kant intended to correct this miscalculated entitlement of the animus and proclaimed that reason and thing are insurmountably divided (Ibid. Cf. Kant 1995). The role of the animus was reduced to normativism and moralism. According to Kant, human cognition amounts to regulation of the rational desire rather than determining it and uniting the person with the beloved. The deep desires of the anima for loving union cannot be articulated and fulfilled through this rationalistic model. Hegel, on the other hand, wanted to integrate the two opposites and devised his giant dialectic to take in everything "in heaven and on earth" (Ibid. Cf. Hegel and der Logik 1832). However, this integration is at the expense of the particular human self.

The models of Kant and Hegel try to alleviate the clash between the human subject and the world. D'Arcy insists, however, that the experience of love necessarily leads one to opposing the world. This is not a psychological effect from the emotions stemming from love. According to d'Arcy, it is engraved in the most fundamental structure of love. This is why the tension stemming from the love-experience is not to be tempered by simply restructuring the agents of love within the soul. It has to do with the restructuring of the self and, indeed, its relation to the world. For d'Arcy it is relating to the world in love that provides the fundamental truth about the self and the world. It is this finding of meaning which he reads into the philosophy of Merleau-Ponty.

It is not until Kierkegaard that post-Medieval Western philosophy realizes that the juxtaposition between the self and the world is foundational and is the real cause of the anxiety of human existence. Being one's own self means being in the world, relating to the world and affirming its reality. The being of the self in the world is characterised by disquietude and the reason for this is the fundamental discrepancy between desire for existence of the loving self and the boundaries of its being. This discrepancy is described differently in the tradition of existentialism: Fear (Angst) in Kierkegaard, Movement in Bergson, Care (Sorge) in Heidegger, etc. For d'Arcy the fundamental reason for the disquietude of the self is love.

In its complexity love is a desire for an all-encompassing existence. However, the being of the self is limited by the boundaries of its nature and its individual existence. A seemingly easy, but in fact erroneous, way out of this discrepancy is the idea that the existence of the human self is necessary. For a finite essence, writes d'Arcy, to think of itself as necessarily alive, would be to make itself God. This would be a sin against the Holy Spirit (D'Arcy 1947, p. 258). The truth is that a human being is out in the world, and one is totally dependent on things that are external to one's own self. The narcissistic illusion of self-sufficiency is dangerous, because it seemingly presents a realization of the desires of the anima for the Absolute. Following the philosopher Guthrie, d'Arcy writes that the false essential Ego, which substitutes for the real existential Ego, is a pseudo-Absolute which cannot guarantee that for which the Self craves, namely self-realization, self-giving and immortality (D'Arcy 1947, p. 263). The real Absolute must not be a replica of the Ego. It should be radically different from it. The true Absolute evokes love, as the most sublime manifestation of the self. "In this act, writes d'Arcy, the will makes haste to aid the intellect and the intellect itself makes an effort to support the will" (D'Arcy 1947, p. 267). The external entity to which love is addressed is seen by the mind in its true worth. Thus, the essential movement towards self-affirmation is not anymore contradictory to the existential

movement away from the self to the Absolute. The pseudo-Absolute of the narcissistic self is transfigured in the rays of the true Absolute (D'Arcy 1947, p. 267).

## 5. The Hierarchy

From the perspective of this realisation of the self in the love for the true Absolute, a hierarchy of being is revealed: "From the moment it exists the Self stretches out with all its antennae for safety and union with the Absolute, and it appraises all the external objects which it meets by the measure of its longing. This primordial movement of the anima expresses itself on the spiritual plane in will and culminates in love." The perceived hierarchy of being corresponds to the hierarchy of the loving self: "The motor-force of the essential self, on the other hand, thrives by appropriating to itself all that it meets. It, too, as we have seen in previous chapters, is primitive and can be brutal and bestial. But it is and must be secondary, since it issues from something which is not yet itself and ends in an Absolute which is finite and relative." (D'Arcy 1947, p. 268).

D'Arcy's view of a sacred order of being recognized in love is congenial to the concept of the hierarchy developed in *Corpus Dionysiacum*—a 6th-century collection of theological and philosophical texts ascribed to the St. Dionysius the Areopagite, the pupil of St. Paul and first bishop of Athens. D'Arcy does indeed mention Dionysius as a thinker who speculated on erotic love (). But apart from this explicit reference there is congeniality between his dialectic of love and Dionysian ideas such as the triplex movement of the sou towards God and the participation of created beings in divine live through the sacred hierarchy. The channels for the influence are not the focus of this article.

The central guarantor for knowledge in the Dionysian model is the hierarchy. The hierarchy is a sacred order, knowledge and revelation; it seeks affinity, as far as possible, to the God-like being, and is led to the illuminations that are obtained analogically and through imitation of God. Thus, according to *Corpus Dionysiacum*, God is beyond everything, being at the same time the source not only of all being, but of all knowledge. Within the mode of divine super-essential existence, God, knowing himself, knows all beings according to their own ontological capacity.

Entering into this hierarchical (hierarchical in the sense of Dionysius) relation to being the self is engaged into three-tier movement. The first two vectors of movement pertain to the self-reflexive aspect of love, whereas the third denotes an extatic movement of the self These are the guarantors for confirming one's own identity(D'Arcy 1947, p. 268). They constitute the motor-force of the self which aims at self-realization.

The first movement of the "essential self" manifests the cognitive perception of all objects of cognition. It is determined by the intentionality of perception and consists in producing images and memories of the cognized objects.

The second movement is the appropriation of the objects met, i.e., the objects of knowledge, to the self. In other words, by structuring the cognized material, the self confirms its own identity. The process at stake is not a purely epistemological one. This is loving knowledge. D'Arcy notes that this movement is primitive and can be brutal and bestial, if it is entirely left over to the anima. If they are controlled by the animus, these two vectors assume their appropriate place as secondary, as "it issues from something which is not yet itself and ends in an Absolute which is finite and relative" (D'Arcy 1947, p. 268). The status the self reaches by appropriating everything it encounters is the result of seeking perfection, but it is human and limited, and therefore, shadowy and uncertain "like the moon which draws light from something else" (D'Arcy 1947, p. 268).

The third movement of the loving self transcends the boundaries of the subject in reaching out to God from whom the existence of the self is derived, and its succour depends. "From the moment it commences to exist", writes d'Arcy, "the Self stretches out with all its antennae for safety and union with the Absolute, and it appraises all the external objects which it meets by the measure of its longing." (D'Arcy 1947, p. 268). The Self stretches to the transcendent source of the visible order of things and adores this source of the Loved One. This movement, underlines d'Arcy, does not neglect the appropriated world. On the

contrary, the appropriated world is intellectually transformed, it is "grasped in ideas of it"; "the macrocosm is reduced to the microcosm of finite mind".

The movement of the loving self described by d'Arcy bears remarkable correspondence to the three-tier movement of the human soul described in *Corpus Dionysiacum* (Areopagita 1990a). While Dionysius is mostly interested in demonstrating the union of soul and body within the process of cognitive illumination and knowledge of the God, d'Arcy takes this union for granted and is focused on the dynamics of personhood. However, the two authors are congenial in that they both consider knowledge not as a merely analytic function, but as a loving relation which is constitutive for the consciousness of the human subject. The other pivotal parallel is that they see the structure of the world as pertinent to the process of loving transfiguration of the self, whereby the difference between the human subject and the world is not blurred in a metaphysic of the personhood of some form or in idealistic solipsism. The structure of the world is relevant to the structure of the self and, hence, it is meaningful in the experience of love.

In Dionysius these three gnoseological paths also require ascetic effort and are determined by the personal will. Nevertheless, in the treatise *De divinis nominibus* the subordination of the cognitive powers is strictly determined by the unifying activity of the intellect. Once activated, the intellect is capable of discerning the truthful knowledge and of achieving it. Through the triple movement of the soul the creative and providential activity of God is manifested and it becomes recognizable.

D'Arcy insists that the ascent of the existential self is beyond the mere contemplation, because the Absolute could be truthfully grasped even by asserting the reality of the finite objects of knowledge. However, the self has a desire to reach to the Absolute in a loving relation, which is beyond mere contemplation. This happens through self-denial and lowliness, being at the same time the highest possible perfection of the self. Again here, and just as in *Corpus Dionysiacum*, we see a combined manifestation of Eros and Agape.

## 6. Excurse: The Phenomenological Approach to the Self-Personhood and the Ontology of Perception

In his monograph *Facing God* d'Arcy criticizes the different ideas of selfhood produced by the tradition of metaphysics. Maurice Merleau-Ponty's ontology of perception seems the most congenial to his own views, with the idea that the world is not incidental to our personal life (D'Arcy 1966). In this context d'Arcy writes: "Merleau-Ponty means that we are bound up through our body with the world around us; it is within its unity that we are free to act and grow. I am, he tells us, an intersubjective field, not despite my body and historical situation; on the contrary, we are born of this world and in this world. For the same reason it is open to us with all its possibilities. Hence we do not merely accept the world; we choose it and give it style and significance" (D'Arcy 1966, p. 47).

Having indicated the conceptual deficit of phenomenology in terms of the analysis of interpersonality, d'Arcy demonstrates that, through love, the profoundly dialogical structure of the person is revealed. The ultimate dialogical partner of each human person is Christ, as he is the most sublime meaning of the reality we live in. The different aspects of perception in love guarantee meaningful transformation of the person. Reality has a structure which is profoundly relevant to the life of the self. D'Arcy is aware of the fact that structure is not identical with meaning but, alluding to Jung's model of the Gestalt, he insists that the two are deeply interconnected when the life of the person is concerned. D'Arcy reconstructs the truthful and meaningful dynamic of the self in view of the reflected experience of the world and the self. For d'Arcy, the epistemological method through which we can reconstruct the structure of the self is derived from the interconnectedness with the world.

The self faces the world in a specific act of loving knowledge—a synthesis, namely, between love and knowledge. On the ground of the synthesis sense-perception gains significance. This requires and encompasses self-denial. This is the impulse "to fly this world and purify soul and body". This transcending of the intersubjective communion with

the world, in which we abide, is a sacramental transfiguration. The tenet of self-denial and transformation makes personal life commensurate with the Christ-event. It is important to note that for d'Arcy self-denial implies transforming nature: "the supernatural life has slowly to dominate and fuse with nature" (D'Arcy 1966, pp. 96–97).

This transformative process introduces change in the personal being. The change has positive meaning, not because it is a completion of the person, or as discovering the authentic personality, as an existentialist conceptualization would sound. At stake is something else. For him the life of the person is rather a process of consolidation than becoming something new. We are all the time complete, underlines d'Arcy. It is not that the person moves from imperfection to perfection. Under consolidation he means confirming one's identity: "Rather we can say together with the theologians that "we can choose to make an act of love or to commit a grievous act and fix our destiny forever" (D'Arcy 1966, p. 98). In that dense, the dynamics of personal life is not about self-improvement; it is projecting eternity within the personal existence.

This consolidation of the person in terms of eternity is not described by d'Arcy as mystical self-knowledge in the traditional sense, i.e., finding in solitude the traces of eternal being within the hidden core of the intellect: *abditum mentis* (Augustine). It is in the bombarding of sensations to which every human being is exposed that said consolidation takes place. Wave after wave, the sensations connect us to the world, and we have no time to attend to each layer of sense perception until the next comes. However, we do not necessarily lose our personal identity within this stream; on the contrary, we strengthen it. In order to illustrate how it is possible to trace a meaning in the stream of sensations, d'Arcy refers to Hegel's model of historical synthesis: "Hegel saw historical occasions as moments in the seeping movement of Universal Spirit, and one wave is subsumed into another wave, helps its impetus, but disappears itself" (D'Arcy 1966, p. 98). D'Arcy transfers this Hegelian model into the realm of the personal human subject. The consolidation of the human person within the dynamic of sensations is possible when perception is confronted with human will[5].

D'Arcy notes that confirming reality in each phase of the personal existence is not a teleological process. It is a dialectic of negation and overcoming. In being negated, each stage of confronting reality becomes meaningful and transformative. In this respect he makes reference to the theory of evolution which introduces a new dimension to change: "Anything can happen in the strange story of man, but in fact all human actions in their right proportion can serve the Eucharistic mystery and enable human beings to give of their substance to the recreation of human life" (D'Arcy 1966, p. 102). The parallel to the Eucharist shows that personal consolidation is an elected mode of existence, rather than a single option of choice. In addition, it can take place at any point on the vector of historical time. The Eucharistic experience and the evolution theory enabled d'Arcy to identify a specific Christian experience of time, opposed to both the cyclical concept of Ancient Greek philosophy and the linear progression envisaged by the religious consciousness of the Jews. Christians, he notes, have an interim understanding: "There then would be a long gap and a long wait. But even here on earth we can almost be in two places at the same time . . . But for all we know, the heavenly society is already being formed or the end of the world has already come" (D'Arcy 1966, p. 106). Our experiences should be transfigured in order to pour into eternity. In addition, it does not matter how far from the end of history this transformation takes place, since everything which happens in time is equidistant from God.

In his book *The Lion and the Unicorn*, d'Arcy conceptualizes the described consolidation of the self in communion with the world using the grammar of love. The different affectations, emotions and existential modes of love are not seen simply as elements of a coherent entity. Rather, these are different existential modes of the loving self, seen as stages in the path of self-giving, as modes of manifestation of the entire person. What should be given up in love is the entire self, not particular elements of it, or some objectified impulses such as *anima* or *animus*. Thus, the discourse of love turns into an existential analytic of the self.

Love is a factor for restructuring and transforming the self which puts the latter into a new disposition to the world. Love is a state of being which opens up the self to the world. In obeying predominantly the anima or the animus, one loses the capacity to loving sacrifice. Obeying each of the two can lead to deterioration of the person.

It follows from this analysis that the truth of authentic being is found in the love which is experienced as a self-consolidation. Love is not a human action, an attitude or an emotion. It is a reality which makes any historical meaning possible. This is possible because of the paradigmatic act of the Incarnation of the Son of God in history. The latter is the most radical expression of divine love, which transforms the human experience of the world. From this perspective d'Arcy points out that the Christian experience provides a completely novel and unique understanding of time. Time in a sense stands still, because it has been fulfilled in Christ and every individual and every generation lives at the end of the world meeting his destiny. According to d'Arcy this has been the unique contribution of Christianity to the philosophy of history, but also to the philosophy of personhood. In light of the incarnation all human achievements, struggles and developments, i.e., the experience of the world, gain a new and everlasting significance (D'Arcy 1966, p. 110). God's love for human-beings enables every single person to come to acceptance, giving up and justification of one's own self.

D'Arcy depicts the layers of personal life on which this loving self-knowledge takes place. These layers include different structures of the human psyche. Foundational for d'Arcy is the differentiation of the male and the female principle, which resonates with the Aristotelian model of the active and passive intellect, as well as with Jung's dichotomy of *animus* and *anima*. D'Arcy gives different definitions of *animus* and *anima* as powers of the self. Quite often he presents the interplay between the active and the passive cognitive desire as a union of mind and will. With the term "mind" d'Arcy denotes what is since John Locke known as *consciousness*. The will is an intellectual appetite according to the model of the act of will of Thomas Aquinas.

"There is no mind without interest", writes d'Arcy, "and no will which is completely blind." It is important to note, that the activity of both the will and the mind is intentional, i. e. it possesses external reference and principle of activation. The mind is always cognizing *something* (I know that . . . ) or is thinking about *something* (I think that . . . ). As far as it is driven to knowledge by the objects, it is passive: it cannot but know the things it encounters. But in another sense it is active: the content perceived by the mind is then contemplated as something known by me. In the same way, volition is always orientated towards *something*. At first glance the will is purely passive, since it is a rational desire towards the object of knowledge; it drives the eye of the intellect to the thing, which is for the will an object of desire. However, following the Medieval Christian anthropology, d'Arcy insists that the will is not merely a reaction to a stimulus; it possess also the power of knowledge and self-reflexivity. The will is not a mere desire towards the object of cognition, but it is a desire to have a desire towards the object of the will. In other words, the act of will is not mere prioritizing of the desires towards different objects of the will. For the will is not a mere ordinative (*ordinativa*) faculty, but a determinative (*determinativa*) one as well. It determines and defines the individual forms of the objects and on the basis of this conceptualization the desires are being ordered and prioritized. Thomas Aquinas constructs a two-tier model of the will: on the one hand, the will is a desire ordering the other desires and, on the other, it is a faculty of cognizing the forms of things. The concrete forms are known through the qualification of the common forms with some individualizing characteristics. Hence, the will is self-reflective, just like the intellect. For d'Arcy this speculation is a proof that every act of the will has a cognitive and a loving aspect. The intentionality of the mind and of the will has a common object which is why these two faculties always perform together in one and the same human act.

On the backdrop of this analysis, d'Arcy demonstrates that the act of love can be seen as a special act of the will. When it comes to love, the self is not and cannot be neutral. Determining the object of love is recognizing and confirming one's own selfhood. Within

love the "passive" act of the will is manifested as profoundly active and decisive. How does this happen? In love the self is confronted with its own creatureliness, and this is a transformative experience. When experiencing love, the person refers to the bodily substrate of the self not as a mere source of perception and desire, but as a base of a much deeper disposition of the personal existence. The person is given a possibility to confirm its existence in a new mode. The perceived creatureliness of human being causes human nature "to hold on existence, to persist in being" (D'Arcy 1966, p. 15). Thus, the lover enters into a special relation to God. D'Arcy quotes Guthrier according to whom humans are "hangers on the Absolute". We receive our being and life from God and this is the most fundamental loving relation, one of giving and taking. The paradigm of this relation is the life of the Trinity itself. As God is the fulfilment of human love, the soul becomes a seeker of divine love. In love the soul discovers its inextricable tendency to seek the heights of the union of love. In that sense all human love is love for God. Love is a communion with the Absolute.

Apparently the "passive" aspect of love, namely the appetite, works as an active power of self-determination in relation to the Absolut, whereas the "active" aspect of love, namely reason, defines one's own passivity in relation to divine love, as the latter is the source and the goal of every human love.

## 7. Recapitulation

D'Arcy uses the phenomenological method of Merleau-Ponty in order to rehabilitate the Eros from its stigmatization by the normative moralism of post-medieval Christian theology (Anders Nygren). He tested the efficacy of this method in explaining the person's existential relation to the world in his book *Facing God*. As a paradigmatic expression of love, the Christ event is constitutive for the manifestation of all forms and impulses of love. Influenced by Maritain's view of perfect love as a full communication of naked selfhood, d'Arcy refuses to see self-sacrificial love as a mask hiding the brutal and possessive impulse of the eros. D'Arcy sees the loving desire with self-sacrificial charity as parameters of the existential situation of man. He refers to modern psychology and uses the psychological paradigms of passivity and self-assertion (*Gestalte*, after Karl Jung) not as explicative mechanisms, but rather as phenomenological markers of the intentionality of the loving self.

With an anti-Hegelian overtone, d'Arcy demonstrates that ecstasy, regress and assertion are not mere moments of the self-reflection of the loving subject; they exemplify the ascent of the entire human being to the core of personal being, to ontological truth and to the dialogue with God. This movement is typologically similar to the ontological and gnoseological dynamics of the Dionysian hierarchy. One of the programmatic ideas in *Corpus Dionysiacum* is that Eros and agape are not mutually exclusive but constitute the loving relation between the highest and the lower levels of being. The participation of the lower levels of being in the divine illuminating activity takes place in relation to the higher ranks of the hierarchy. The process is revelatory and transformative. D'arcy's solution is closer to this model rather than to the neo-Thomistic model of the analogy of created and divine being. One has to bear in mind that Thomas Aquinas himself diverges from the *analogia entis*, when it comes to expressing of the ontological pillar of personal being. He makes use of Pseudo-Dionysius.

For example, moral theology is loaded with criticism against the Dionysian view of the erotic. Dionysius has been accused of introducing pagan intuitions into the concept of Christian love. D'Arcy's scope is to rehabilitate the eros - not in a narrowly doctrinal or strictly philosophical aspect, but as a manifestation of the dynamic of the self. For d'Arcy the erotic has a meaning, much deeper than the function of representing desires, needs or aspirations. The erotic is not mere expression of utilitarian needs and desires. If the eros is a revelation of the intentionality of personal life, the manifestations of the erotic should be seen as images pointing to an end and a task, but at the same time bringing joy and pleasure.

This speculation brings d'Arcy to the problem of the ontological and epistemological functions of the image as a symbol of human love. Thus, the ancient antinomy between the loving desire (eros) and the self-sacrificial charity (agape) is resolved, as the two are seen as characteristics of different levels in which images function. This meaning of the image is calibrated in correspondence with their function to evoke intention. The intentional interpretation of the image is one of the pillars of the aesthetic theory of Thomas Aquians, indebted to his reading of Aristotle, but also to his analysis of the image concept of Pseudo-Dionysius. In fact, the intentional function of images in Dionysius has been the focus of authors (philosophers and aestheticians) who have contributed significantly to the modern aesthetic theory of the symbol.

Although d'Arcy is influenced by Maritain's view of perfect love as a full communication of the naked selfhood, his concept surpasses the theology of neo-Thomism, as it does not rely on the analogy of created and divine being. D'Arcy is much closer to the programmatic Dionysian model in which eros and agape—seen not as mutually exclusive, but rather as constitutive of a loving relation between different types of images—guarantee the participation of created being in the revelatory presence of God. The order, scope and the transformative capacity of the participation follow the pattern of the celestial and the ecclesial hierarchy.

This dialectic corresponds to the order of reality. And the symbolic knowledge is the transcendental base which co-ordinates the dialectic of love and the structure of reality. But how are symbols themselves validated? The answer is: *through their capacity to express the love and the existence of the self*.

**Funding:** This research received no external funding.

**Conflicts of Interest:** The author declares no conflict of interest.

## Notes

[1]    Daniel Day Williams, *Op. cit.*: "This mystery must be traced down into the existence of two loves within man. D'Arcy develops the doctrine of the two loves by identifying eros as belonging to the essential self. This love seeks fulfilment. It is possessive, masculine, imperious and it denies the completion of personal being. It dominates the rational impulses and the will to understand. The other love is identified with the existential self. It is the love which seeks to give itself away. It is emotionally powerful in its heedlessness. It is feminine, intuitive and spendthrift. It is the agape in the self . . . ".

[2]    Daniel Day Williams, *Op. cit.*

[3]    Daniel Day Williams, *Op. cit*. Cf. (D'Arcy 1947, p. 321).

[4]    Plato, *Republic* bk. 9, 589A6-B6.

[5]    (D'Arcy 1966, p. 99). In this sense d'Arcy references to a distinction made by Aristotle. The latter differentiates processes which have no significance or value, except in the result, and processes which are confirmed by the will at every stage of their unfolding. Thus, for instance, a dish maybe initially quite unpleasant to look at, but in the end it is a delight. Aristotle calls these actions imperfect. In contradiction to such processes, the acts of the will are perfect and complete (τελείαι). The volitional confronting with reality of life transform each moment of sense perception into an act of committing oneself to a meaning. This committing transforms our experience of time making it kairos (καιρός, i.e., meaningful temporality), whereby the cognitive processing of sense data gains the significance of decision—krisis (κρίσις).

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
