# Peer review of "Love as Consolidation of the Self-in-the-World: Martin d’Arcy’s Speculation on Love as a Metaphysical Supplement to Phenomenology"

_religions, doi:10.3390/rel14040454_

Round 1

Reviewer 1 Report

There are some really interesting and insightful discussions presented here, however I don't really get a sense of the 'problematic' or 'hook' for the discussion until line 395 onwards - I think the introduction could do far more in making clear and explicit a focus/rationale to the argument that guides the reader through the proceeding discussion. If indeed misapprehensions about Eros and the erotic are what animates the discussion, this section perhaps even needs to be relocated to the introduction (or at least more closely mirrored so that introduction and conclusion are more clearly linked) so that this is clear and understandable for the reader. I'm also not sure if the last paragraph fully encapsulates and synthesises everything that the article is trying to explore or do - but that might be to do with my misapprehensions and a lack of clarity on that point throughout the discussion. 

There is also no bibliogrpaphy, which makes it more difficult to comment on the references to contemporary scholarship at a broader level. Attentiveness to d'Arcy and his context strikes me as excellent and a notable strength of the article. 

More detailed points below:

Line 56 - delete 'the' 

Lines 56-7 - scope to make clearer the focus/aims of the article? In particular, what it seeks to establish and why that is significant for readings of d'Arcy but also broader theological notions of love (vis-à-vis theological notions of selfhood)

Line 61 - sp. Maurice

Line 63 - make clearer 'he' refers to d'Arcy

Line 95 - dash should be a comma

Line 235 (and elsewhere) - why is d'Arcy's forename repeated every so often?

Line 279 - Descartes' (misplaced apostrophe)

Line 282 - delete 'the'

Line 296 (and 395) - sp 'Merleau-Ponty'

Line 415 - capitalisation of 'D'Arcy' to be addressed

Author Response

Dear Reviewer,

Heartfelt thanks for your remarks and recommendations! I have done some editing and have introduced some corrections!

Author

Reviewer 2 Report

The stated goal of the theological and philosophical research in this article is to introduce Martin d’Arcy’s key concept of Christian love by investigating book The Mind and Heart of Love. The topic is very interesting, but I have some concerns about the manuscript and these are listed below.

1. This manuscript lacks the section of literature review in its Introduction. At the start of the essay, the author should synthesize the existing literature on the topic and questions into different theoretical perspectives from which the conceptual framework (maybe the ‘phenomenological model’?) is derived and developed. That is to say, the literature review of the existing publications should be clearly and closely relevant to the research and explains the basis for the assumptions to be investigated. Literature review is NOT a background introduction or present what is reported or known. If author read some well written research papers published, you should know how to use those papers as a template to develop a paper that would follow the logic of research design and the standard template and format. In any way, the paper should be revised to meet the above standard.

2. The paper is far too descriptive and fails to adopt a critical reading. The author has the main body, but simply needs an explicit conceptual section that would clarify your phenomenological model to begin with your research into the theological research and address the methodological novelty under study.

3. It seems that the comparative research (“The Hierarchy”, from line 324) on d’Arcy and Pseudo-Dionysius is not systematic enough. Owing to the comparative nature of such a research, the discussion on both two philosophers becomes sporadic and insufficient. Although considering the spatial limits of the work, the author could further discuss the theology of Pseudo-Dionysius from a modern phenomenological perspective.

4. Some of the author’s representation of some philosophers are not strictly based on the textual evidence in their works. For instance, the author states, “According to Kant, the human cognition amounts to regulation of the rational desire rather than determining it and uniting the person with the beloved. The deep desires of the anima for loving union cannot be articulated and fulfilled through this rationalistic mode. Hegel, on the other hand, wanted to integrate the two opposites and devised his giant dialectic to take in everything ‘in heaven and on earth’.” (Line 282-286) However, the author does not mention which quote from Kant and Hegel can support this interpretation.

5. The following expression is unclear and misguiding: “While Dionysius is mostly interested in demonstrating the union of soul and body within the process of cognitive illumination and knowledge of the God, d’Arcy takes this union for granted and is focused on the dynamics of personhood. But the two authors are congenial in that they both consider knowledge not as a merely analytic function, but as a loving relation which is constitutive for the consciousness of the human subject.” (Line 372-376).

6. The author needs to add more in-text citations and provide more references to primary and secondary literature.

Author Response

(The authors gave the same response as above.)

Reviewer 3 Report

This is a highly original and compelling discussion of divine and human love that neatly and productively puts a modern theologian and philosopher into dialogue with a core element in early and eastern Christian spiritual tradition. 

Author Response

(The authors gave the same response as above.)

Round 2

Reviewer 2 Report

I'm satisfied with the author's revisions.  I recommend this paper for publication.